# Gangliosides and Their Role in Multilineage Differentiation of Mesenchymal Stem Cells

**DOI:** 10.3390/biomedicines10123112

**Published:** 2022-12-02

**Authors:** Francesca Santilli, Jessica Fabrizi, Fanny Pulcini, Costantino Santacroce, Maurizio Sorice, Simona Delle Monache, Vincenzo Mattei

**Affiliations:** 1Biomedicine and Advanced Technologies Rieti Center, Sabina Universitas, Angelo Maria Ricci 35A, 02100 Rieti, Italy; 2Department of Experimental Medicine, Sapienza University, Regina Elena 324, 00161 Rome, Italy; 3Department of Biotechnological and Applied Clinical Sciences, University of L’Aquila, Vetoio, 67100 L’Aquila, Italy

**Keywords:** gangliosides, umbilical cord derived stem cells, bone marrow derived stem cells, dental pulp derived stem cells, adipose derived stem cells, mesenchymal stem cells

## Abstract

Gangliosides (GGs) are a glycolipid class present on Mesenchymal Stem Cells (MSCs) surfaces with a critical appearance role in stem cell differentiation, even though their mechanistic role in signaling and differentiation remains largely unknown. This review aims to carry out a critical analysis of the predictive role of gangliosides as specific markers of the cellular state of undifferentiated and differentiated MSCs, towards the osteogenic, chondrogenic, neurogenic, and adipogenic lineage. For this reason, we analyzed the role of GGs during multilineage differentiation processes of several types of MSCs such as Umbilical Cord-derived MSCs (UC-MSCs), Bone Marrow-derived MSCs (BM-MSCs), Dental Pulp derived MSCs (DPSCs), and Adipose derived MSCs (ADSCs). Moreover, we examined the possible role of GGs as specific cell surface markers to identify or isolate specific stem cell isotypes and their potential use as additional markers for quality control of cell-based therapies.

## 1. Introduction

Stem cells (SCs) can be classified based on several criteria like the differentiative capacity (totipotent, pluripotent, multipotent, etc.) or the source of derivation (embryos or adult tissues). Embryonic stem cells (ESCs) are pluripotent cells derived from the inner cell mass of the blastocyst (ICM). Once extracted, these cells can be cultured and proliferate as undifferentiated lines or they can differentiate into desired cell lines [1,2]. 

MSCs are part of adult stem cells (ASCs), non-specialized cells that reproduce to provide some specific cytotypes. The primary function of ASCs is to act as a reservoir of new cellular elements of a given tissue, providing for its structural maintenance and repair of any losses due to trauma or pathologies. MSCs were discovered and characterized for the first time in the bone marrow stroma and called BM-MSCs [3]. Several studies have shown the presence of these cells not only in bone marrow (BM) but also in adipose tissue (AT), umbilical cord (UC), dental pulp (DP), and other tissues [4,5]. MSCs have a degree of plasticity higher than other types of ASCs and can differentiate in vitro in different cell populations; for this reason, they are considered promising in regenerative-reparative medicine, cell therapy, and tissue engineering [6,7].

GGs are the main membrane components expressed on the external cell surface in direct contact with the extracellular environment [8]. GGs are glycosphingolipids (GSLs) complexes consisting of ceramide and a bulky sugar chain containing one or more sialic acids [9,10,11] (Figure 1A). They are also involved in the formation of lipid rafts [10,12,13]. Lipid rafts are highly dynamic structures envisaged as lateral assemblies of specific lipids and proteins in cellular membranes involved in several cellular processes [11,14] such as membrane transport, viruses’ entry, signal transduction, cell adhesion, migration, or apoptosis [15,16,17].

In recent years, several scientists showed the switch of GGs pattern during MSCs’ differentiation, suggesting a possible role of GGs in the mechanism of stem cells’ differentiation. Therefore, GGs are gaining increasing attention in the stem biology field (Table 1). Several specific ganglioside markers have been identified in SCs [18,19]. For example, SSEA-4 (a globo-series ganglioside having a NeuAcα2–3Galβ1–3GalNAcβ1-R structure) [20] is specifically expressed in human pluripotent embryonic stem cells (hESCs) [1] and in induced pluripotent stem cells (hiPSCs) [21,22].

Furthermore, a comparison of the glycome, including glycosphingolipids of hESCs and hiPSCs has also been reported [23]. Overall, these studies have demonstrated that carbohydrate markers can be used as important tools to confirm the pluripotency, quality, identity, and safety of pluripotent cell lines.

GD3 is expressed in mouse and human neural stem cells [24,25], and GD2 and SSEA-4 are expressed in human MSCs [25,26,27]. Moreover, it was shown that GGs play a fundamental role both in mouse embryonic stem cells (mESCs) and in DPSCs’ neuronal differentiation process [28,29].

Martinez et al., intriguingly, demonstrated the expression of GD2 only by MSCs and no other cells within the bone marrow (hematopoietic cells), suggesting that this antigen may represent a single definitive marker of marrow-derived MSCs [26].

GGs were first isolated from ganglion cells by Ernst Klenk in 1942 [30,31]. They are GSLs, composed of an amphiphilic molecule consisting of a hydrophobic ceramide tail which is usually anchored to the outer leaflet of the plasma membrane, and a glycan headgroup represented by a bulky chain sugar containing one or more sialic acids (Figure 1A) [9,10,11]. Currently, the gangliosides nomenclature proposed by Svennerholm in the 1960s remains widely used [32]. According to this nomenclature, GGs are named and classified following the number of sialic acid residues and fitting to their chromatographic mobility. GGs are abbreviated with two letters followed by a number, e.g., in GM3 the G refers to the ganglio-series of glycosphingolipids; the M stands for ‘mono’ which means 1 sialic acid, and the number 3 is based on the migration order in thin layer chromatography (TLC) (Figure 1B) [33].

GGs have important biological characteristics and functions [34], including regulation of cell proliferation, adhesion, migration, apoptosis, cell-cell interactions, cell differentiation [35,36,37], and interaction with proteins [38,39].

Moreover, GGs have been implicated in numerous pathophysiological processes, such as human malignancies [40]. One hallmark of cancer is the ability of cancer cells to undergo epithelial-to-mesenchymal transition (EMT). Through this cellular process, cancer cells lose their epithelial features to acquire mesenchymal features, becoming more aggressive and resistant to drugs [41]. Liang et al. demonstrated an association between GSLs alteration and cancer EMT [42]. During the EMT transition, they found an increase in ganglioside expression accompanied by a drastic reduction of globosides, suggesting that the last one is involved in the maintenance of epithelial characteristics [43]. Furthermore, the dynamic balance between gangliosides and globosides has been reported in mesenchymal-type pancreatic cancer cells compared to their epithelial counterparts [44], suggesting once again the involvement of GSLs during EMT.

In this review, we focused our attention on MSCs because these cells have fewer restrictions than ESCs despite showing great potential in regenerative medicine. Some authors have reported that GGs are important for modulating cell proliferation and neuronal and osteoblast differentiation of MSCs [18,45,46]. They also observed a change in ganglioside expression patterns in cells during differentiation and in response to cytokine and growth factor exposure [47,48]. Numerous studies have confirmed that GGs and their expression levels are controlled during development [49] and are cell type-specific [50], supporting the idea that these molecules are key players in cell commitment [51]. GGs could also play an important role in DPSCs’ differentiation [45,52,53], and our previous studies have shown how specifically lipid rafts are involved in DPSCs’ neuronal differentiation process [4,5,53].

For this reason, the aim of this review is to summarize the role of GGs during the multilineage differentiation processes of MSCs. In addition, since gangliosides localize on the cell surface, we would examine the possible function of GGs as specific cell surface markers to identify or isolate specific stem cell clones and their potential use as additional markers for evaluation of state and quality control of cell-based therapy products [54].

## 2. Main MSCs’ Properties

MSCs are undifferentiated cells relatively easy to isolate from different sources such as BM, umbilical cord blood (UCB), AT, placenta, and dental tissues, and they grow well in culture [67]. MSCs are multilineage cells able to self-renew and differentiate in multiple cell types, which play prominent roles in tissue healing and regenerative medicine. In response to some stimuli such as inflammation, trauma, and necrosis, MSCs can proliferate in vitro and can initiate the differentiative process for repairing damaged tissues in various degenerative diseases, both in animal models and in human clinical trials [68,69,70]. MSCs’ homing ability confers them the capacity to differentiate into local components of injured sites. They are also able to secrete chemokines, cytokines, and growth factors that help tissue regeneration [71,72]. The use of MSCs may reduce some of the controversial ethical issues and technical problems associated with ESCs’ use, as it is harvested from human embryos before preimplantation [73]. MSCs have a distinct morphology and express a characteristic set of surface markers, such as clusters of differentiation (CD) [74,75,76,77,78]. The biological characteristics of MSCs depend on the sources currently used for their isolation. Flow cytometry can be used to define and validate either homogeneity or heterogeneity (i.e., potential contamination) of MSCs in a tissue-specific way and to study their differentiation potential depending on the tissue origin. 

## 3. Role of Gangliosides during Multilineage Differentiation of MSCs Isolated from Different Sources

### 3.1. Umbilical Cord Derived Mesenchymal Stem Cells (UC-MSCs)

The UC is a deciduous and therefore temporary, anatomical formation containing the blood vessels that connect the fetus to the placenta. For many years the UC has been treated as a waste material. In the last decade, however, it has been reconsidered and used for biomedical products and in regenerative medicine, thanks to the presence of MSCs obtained directly from the umbilical cord in toto or from its compartments (amniotic epithelium, sub-amniotic stroma, intravascular stroma named classically as Wharton’s jelly, perivascular stroma, and vessels) [79,80].

UC-MSCs are adherent to plastic and displayed fibroblastic morphology. UC-MSCs have immunomodulatory properties such as the ability to inhibit the proliferation of T, B, and NK lymphocytes and reduce inflammation by secreting interleukin-10 (IL-10) and interleukin-4 (IL-4). Also, they have anti-inflammatory effects, such as suppressing the secretion of inflammatory factor interleukin-1β (IL-1β), tumor necrosis factor-α (TNF-α), and interleukin-8 (IL-8). Next to the immunomodulatory properties, UC-MSCs have a secretory and paracrine activity important in regenerative medicine as they secrete soluble molecules such as keratinocyte growth factor (KGF), hepatocyte growth factor (HGF), epidermal growth factor (EGF), and other cytokines [81,82]. UC-MSCs have a high proliferative potential and capability to differentiate into the three germ layers: mesoderm (adipocytes, osteocytes, and chondrocytes); ectoderm (neurons, astrocytes, and glial cells), and endoderm (islet cells and liver) [83]. Cell differentiation is a highly regulated process that depends on various extracellular and intracellular factors of its modulation, including GGs. Several authors have demonstrated that GGs, such as GM3, GM1, and GD2, are already present in undifferentiated UC-MSCs [84]. 

Nan et al. demonstrated that GM1 induces the differentiation of UC-MSCs into neuron-like cells in vitro, characterized by the expression of the neuron-specific proteins, microtubule-associated protein 2 (MAP-2), and neurofilament protein (NFH), but not glial fibrillary acidic protein (GFAP), a marker for astrocyte development. As UC-MSCs are sub-totipotent stem cells, GM1 may provide a microenvironment to activate the specific expression programs of nerve cells and thereby induce them to differentiate into neural cells [85,86].

Jin et al. were the first to show the presence of GD2 ganglioside in the membrane and to suggest that GD2 expression is closely associated with neuronal differentiation of human UC-MSCs. GD2 can be used as a marker of neuronal differentiation as its expression is also related to the upregulation of early neurogenic transcriptional factors [84]. 

Xu et al. have found that UC-MSCs express GD2 in early passages, and they are the only ones in the chord that express this marker. Their data show that GD2-expressing cells are a subpopulation of primitive precursor cells and suggest that these GGs may be used as markers for isolation in the early culture steps. Furthermore, their study demonstrated that undifferentiated UC-MSCs expressed GD2, as well as differentiated cells, and GD2^+^ MSCs subpopulation which also showed a large upregulation of specific adipogenic, osteogenic, and neurogenic genes, suggesting the involvement of GD2 in differentiation (Table 2) [80].

### 3.2. Bone Marrow-Derived Mesenchymal Stem Cells (BM-MSCs)

BM-MSCs are adult, multipotent, non-hematopoietic stem cells located in the bone marrow stroma, which can be readily harvested and isolated in humans from the sternum, vertebral body, iliac crest, and femoral shaft [93,94,95]. Today, the most common source of MSCs is BM, but bone marrow from healthy donors is a harmful and painful source. BM-MSCs are fibroblast-like cells with colony-forming and multilineage differentiation capabilities [96,97]. These are a population of cells with different commitments [98] and the multipotency capacity to differentiate into osteoblasts, chondrocytes, and adipocytes, as well as into cardiomyocytes, skeletal muscle, and neural precursors [99,100,101,102]. Among ASCs, BM-MSCs are the most studied, and due to their characteristics, such as multipotent differentiation potential, myelo-supportive capacity, anti-inflammatory and immunomodulatory properties, are used in cell therapy and tissue repair [103,104,105,106]. In response to injury signals, BM-MSCs can potentially move from their niche into the peripheral circulation and reach target tissues through vessel walls [104,107]. Gangliosides, GM3, GM2, GM1, GD3, GD2, and GD1a, are usually expressed on the cell surface of undifferentiated cells but also during the differentiation processes in which they seem to be involved [26,52]. During BM-MSCs’ osteogenic differentiation, a drastic decrease in GM3 and GD3 concentration has been demonstrated. In parallel, GM2, GM1, and GD1a increase, and GD1a becomes the ganglioside with the highest relative increase in expression during the differentiation process [29], thus demonstrating that the increase of GD1a ganglioside is crucial for BM-MSCs differentiation [52]. On the other hand, after the chondrogenic differentiation of BM-MSCs, the expression of GM3 temporarily increases. Ryu et al. reported that GM3 and GD3 are expressed after the chondrogenic differentiation and GM3 enhanced transforming growth factor-beta (TGF-β) signaling via SMAD 2/3 [37]. However, GM3 levels significantly decreased, whereas GM1, GD3, and GD1a levels increased during further differentiation into chondrocytes [87]. Furthermore, GM3 is related to the differentiation of megakaryocytes, CD4+ T cells, CD8+ T cells, osteoblasts, and neural cells (Table 2) [28,35,47,88,89].

### 3.3. Dental Pulp Mesenchymal Stem Cells (DPSCs)

DPSCs are multipotent stem cells [108,109] from highly vascularized connective tissue located in the center of the cavity of permanent third molars or in other dental tissues, such as the periodontal ligament, the gingival mucosa, the apical papilla, the dental follicle, or the dental pulp of childhood deciduous teeth [110] that show the properties of MSCs [111,112,113]. DPSCs were initially isolated and characterized by Gronthos et al. [114] and, in recent decades, they became the most studied type of dental stem cells, thanks to their easy extraction, absence of ethical issues, and relative abundance as biological waste from dental clinics [115]. Several researchers reviewed the isolation procedures, characterization, differentiation, and banking activity of DPSCs, providing non-invasive techniques and it has no notable ethical constraints [116,117]. DPSCs exhibit promising characteristics such as BM-MSCs, including fibroblast-like structure, clonogenicity, plasticity, rapid proliferation, regeneration, high proliferative, self-renewal, and multilineage differentiation ability [109,118]. They have the potential to differentiate into endodermal (respiratory and gastrointestinal tracts, liver, pancreas, thyroid, prostate, and bladder lineages), mesodermal (adipogenic, osteogenic, and chondrogenic lineages), and ectodermal (skin and neural lineages) [119,120] when placed in specific culture conditions. Several recent reviews have documented the current knowledge and understanding of DPSCs’ differentiation into vital lines, including their angiogenic, neurogenic, odontogenic, and chondrogenic potential and regeneration of periodontal and dental tissues [121,122,123]. Several studies have shown that GM3, GM2, and GD1a are expressed on the cell surface of undifferentiated DPSCs [36,45]. Therefore, the roles of GGs in differentiation depend on the origin of the MSCs. Lee et al. compared ganglioside expression for human adipose mesenchymal stem cells (ADSCs) and DPSCs differentiation into osteoblasts cells. They found that during DP-MSCs’ osteogenic differentiation, GM3, GM2, and GD1a were mostly increased [36,45] with a significant increase in GD1a expression compared to ADSCs [39]. Previous studies suggest that ganglioside GD1a is essential in regulating the differentiation of MSCs into osteoblasts [28,35]. Moussavou et al. suggest that GGs might play a role in the differentiation of ADSCs and DPSCs into osteoblasts and that this role is more important in regulating the osteoblast differentiation of DP-MSCs compared to ADSCs [36]. Immunostaining and high-performance thin-layer chromatography (HPTLC) analysis showed that an increase in ganglioside biosynthesis was associated with the neural differentiation of DPSCs. In fact, during DPSCs’ neuronal differentiation, GM3, GD3, and GD1a are expressed, with a significant increase in GD3 and GD1a expression [29] and furthermore, Martellucci et al., have shown that GM2 was the most representative ganglioside in DPSCs whilst GD3 was present exclusively during DPSCs’ neuronal differentiation (Table 2) [4,5]. 

### 3.4. Adipose Mesenchymal Derived Stem Cells (ADSCs)

AT is derived from the mesoderm during embryonic development, and it is present in every mammalian species, located throughout the body. AT serves as an endocrine organ, functioning to maintain energy metabolism through the storage of lipids [124]. The long-held belief about AT was that it was a relatively inert tissue in terms of biological activity, and it was believed that its main role was energy storage. However, this old theory was changed thanks to the discovery of the large abundance of adult stem/stromal cells in this tissue. ADSCs are multipotent stem cells and the actual progenitors of fat cells/adipocytes [125]. Today, AT is the predominant cell source for MSCs [126] which can be obtained more easily and efficiently than from BM [127]. The most common source of ADSCs is the white adipose tissue present in subcutaneous tissues, in the intraperitoneal compartment (visceral fat surrounding organs), and spread throughout the body as a padding for vital structures. SCs are principally isolated from adipose tissue of the abdomen, thighs, and hips/buttocks by physical methods of enzymatic digestion, typically with collagenase [128]. The resulting cellular pellet is indicated as the stromal-vascular fraction (SVF) and it is composed of all cell types present in AT, such as endothelial cells, resident, or infiltrated macrophages, pericytes, lymphocytes, and adipocyte precursors. ADSCs are selected from the other SVF cells by their adhesion on culture plates; they are a relatively homogeneous population of spindle-shaped fibroblast-like cells that expand after 7–14 days of culture [129,130]. ADSCs constitute a communication network that regulates the activity and function of adipose tissue deposits. ADSCs can interact with other cells and modulate important processes, including inflammation, apoptosis, and angiogenesis [131]. They can regulate angiogenesis through paracrine mechanisms by releasing growth factors such as vascular endothelial growth factor (VEGF), HGF, and basic fibroblast growth factor (bFGF). ADSCs have immune-modulatory activity through the secretion of IL-6 and TGF-β1 and release other important factors such as IGF-1, IL-8, and BDNF by promoting their application in tissue engineering and regenerative medicine [132]. When placed in contact with specific growth factors, ADSCs can differentiate into many different cell types of the mesodermal (bone, fat, cartilage, cardiac, and muscle cells) and non-mesodermal lineages (neuron-like cells, endothelial cells, hepatocytes, pancreatic cells) [133]. Many studies have shown that GGs play a crucial role in the differentiation of MSCs [85] and were differentially expressed during neural and osteogenic differentiation of MSCs [36,51,52,55]. GM3, GM2, and GD1a gangliosides are also expressed in undifferentiated ADSCs, and GD1a expression increased during differentiation towards an osteoblastic phenotype [36,52]. The ganglioside GD3 is known to be involved in neurogenesis and it is also considered important in the maintenance and proliferation of Neural Stem Cells (NSCs). Cho et al. demonstrated the critical role of ganglioside GD3 in the neuronal differentiation of pig adipose stem cells. Specifically, enzyme ganglioside synthase inhibition causes the blocking of ganglioside GD3 expression followed by reduced neuronal differentiation of ADSCs [90]. Regarding the adipogenic differentiation of MSCs, GM3 is the highest in adipocytes. Rampler et al. showed a significant increase in GM3, which is known to be expressed in adipocytes differentiated from adipose tissue [91,92]. In addition to these studies, Hohenwallner et al., recently found that GGs are upregulated in adipocytes compared to their human MSC progenitors, and confirmed the presence of these classes (GM3, GM2, GM1, GD3, GD2, and GD1a) in undifferentiated MSCs. They found that GM1 is abundant during adipogenic differentiation, it increases in osteogenic and decreases in chondrogenic differentiation. GM3 is expressed more in adipogenic than in osteogenic differentiated ADSCs. GD2 is not expressed in chondrogenic or osteogenic cells, but in ADSCs and differentiated adipogenic cells (Table 2) [54]. 

## 4. Gangliosides as a Potential New Class of Stem Cell Markers

The expressions of GGs are frequently and drastically changed during development and differentiation; therefore, GGs can be useful as lineage-specific differentiation markers for identifying or isolating stem cells [18,19]. Recently, attention has been placed on GGs that appear to act as molecular markers for the recognition of multipotent stem cell subclasses (Figure 2). For example, GM1, GM3, and GD2 were expressed in UC-MSCs and BM-MSCs [26,84]. This ability depends on the fact that they can be addressed by specific monoclonal antibodies, thus providing a tool for cell recognition and separation [18,134]. GGs expressed in several stem cells (pluripotent, multipotent, and cancer) have been identified by biochemical and immunological analysis [55,84,135]. Several studies have hypothesized that GGs could be used as specific markers for cell lineage or cell status [36,136]. It is currently known that GGs were differentially expressed during neural [55] and osteogenic [36,52] differentiation of MSCs. Little is known about the other lineage differentiation processes. Ganglioside GD2 may serve as a marker for the identification and purification of murine bone marrow mesenchymal stem cells (mBM-MSCs) [137]. Moreover, Xu et al., demonstrate that the cells selected by GD2 are a subpopulation of MSCs with primitive precursor cells features and provide evidence that GD2 can be considered a cell surface marker suitable for the isolation and purification of UC-MSCs in early-passage culture [80]. Meanwhile, Martinez et al., confirmed that cells selected by GD2 are a subpopulation of MSCs with primitive precursor cells features and that GD2 is highly expressed in BM-MSCs, therefore, this marker is being used for the prospective isolation of these cells [26]. Bergante et al. assessed that ganglioside GD1a could play a role in BM-MSCs differentiation, immunomagnetically sorted BM-MSCs with an anti-GD1a antibody, obtaining a cell fraction highly positive for GD1a (GD1a^+^) and a cell fraction poorly positive for GD1a (GD1a^−^). GD1a^+^ cell fraction revealed a significantly higher expression of osteogenic markers than GD1a^−^ BM-MSCs, confirming their higher commitment toward the osteogenic phenotype. These results support the hypothesis that GGs could potentially be used not only as stem cell markers but also to direct stem cell differentiation [52]. Nakatani et al. examined the expression of GD3 in NSCs and evaluated the usefulness of GD3 as a cell-surface biomarker for identifying and isolating NSCs, especially from postnatal and adult brain tissue in which GD3 is rare but GM1 and GD1a are abundantly expressed. They found that in embryonic, postnatal, and adult NSCs, the percentage of GD3^+^ cells were more than 80% and possessed all the characteristics of NSCs, such as marker expression (nestin, Sox2, Musashi-1, and SSEA-1), a neurosphere-forming ability, and a multipotency to differentiate into neurons, astrocytes, and oligodendrocytes [24]. Wang et al. suggest that GM1 and neural growth factor (NGF) induce NSCs proliferation and differentiation, respectively, in a dose-dependent manner [138]. Rossdam et al., for the first time, describe GM3 and GD3 as being highly abundant GSLs on the cell surface of stem cell-derived cardiomyocytes [139]. Rampler et al. propose several GM3 ganglioside species as potential markers for the characterization of differentiated adipogenic cells and found that there are other classes of ganglioside including GM4 and GT3 significantly upregulated in the chondrogenic lineage compared to native MSCs [92]. Battula et al. report that ganglioside GD2 identifies a small fraction of cells in human breast cancer cell lines. Their findings suggest that GD2 is a possible novel biomarker of breast cancer stem cells (CSCs) and that GD3S, the enzyme involved in GD2 biosynthesis, is essential for breast CSCs function [140,141]. Also, in the brain, a subpopulation of brain cancer cells has been reported. These cells exhibit stem cell-like characteristics, such as the ability for self-renewal and multipotency, in addition to the capability to sustain brain tumor formation. These cells also express c-series gangliosides, also known as A2B5 antigens, characteristic of embryonic cells [142,143]. These GGs can be utilized not only as biomarkers for CSCs but also as targets for the treatment of tumors [144,145]. Considering this, we hypothesize that studies of stem cell gangliosides and their role in stem cell differentiation and other processes should prove to be a fertile area of research in the future. 

## 5. Conclusions and Future Perspectives

In this review, we analyzed the role of GGs during MSCs’ multilineage differentiation process and the possibility of using GGs as specific markers to isolate MSC clones and maintain their cellular identity and functionalities during ex vivo culture in therapeutic applications. Several authors have shown a switch in the gangliosides pattern during the multilineage differentiation process of MSCs. Indeed, GGs’ expression frequently and drastically changed during development and differentiation, so GGs have been discussed as possible stem cell and lineage-specific differentiation markers. As for hiPSCs, their identification may contribute to the knowledge about their role in several biological processes including maintenance, proliferation, and differentiation. Hence the identification of expression of specific GGs at a different stage of MSC commitment may enable the use of these molecules as a potential target to isolate specific MSC clones (magnetic immunodetection) or as novel glycolipid-based CD markers for stem cell phenotype characterization and targeting of human stem cell clones. In conclusion, a better knowledge of gangliosides expression in stem cells will definitely allow a better separation of specific clones for their utilization in regenerative medicine. Moreover, as suggested for human embryonic and induced pluripotent stem cells [146], MSCs represent also a new tool of regenerative medicine to study MSCs-based cell therapies for disease treatments. It is noted that one of the main limitations of this work was the poor and little detailed scientific literature on this subject which made it difficult having a conclusive overview of the gangliosides pattern variation during MSCs differentiation processes.

## Figures and Tables

**Figure 1 biomedicines-10-03112-f001:**
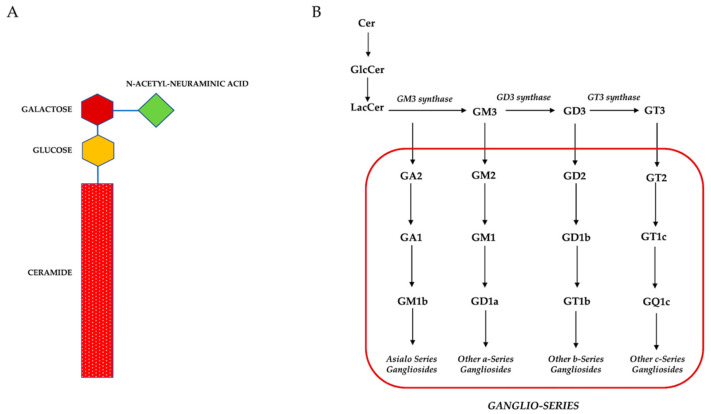
(**A**): Schematic representation of gangliosides structure (i.e., GM3). (**B**): Schematic representation of gangliosides metabolism.

**Figure 2 biomedicines-10-03112-f002:**
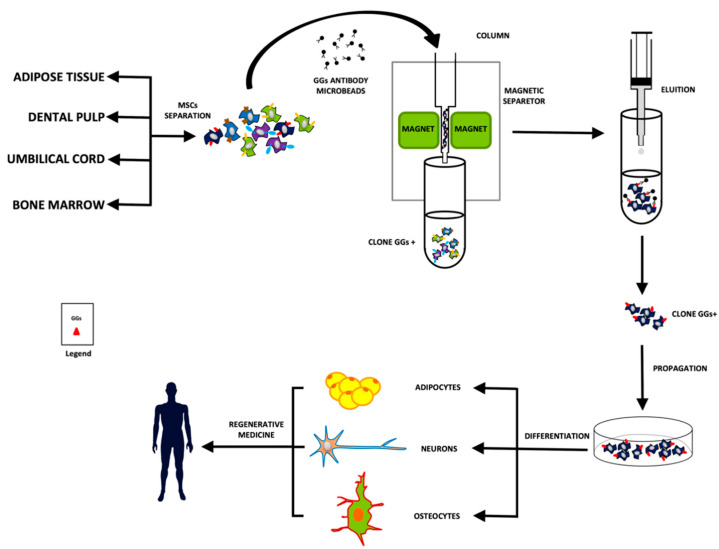
Schematic procedure for the separation of the specific clones of MSCs.

**Table 1 biomedicines-10-03112-t001:** Gangliosides and their role in stem cells.

Gangliosides	Role in Stem Cells	Ref
**GM3**	▪ Regulates cell differentiation and proliferation. ▪ Regulates the initiation step of osteoblast differentiation.	[47,55]
**GM2**	▪ Is the most representative ganglioside on the plasma membrane of DP-MSCs and associates with cellular prion protein (PrP^C^) by EGF receptor (EGF-R).	[4,5,29]
**GM1**	▪ Mediates the pathophysiology of several cellular processes, including development, differentiation, and protection of neuronal tissues. ▪ Regulatory role during neurogenesis and regeneration. ▪ Promotes the proliferation of neural stem cells.	[56,57]
**GD3**	▪ Is the major ganglioside species in neural stem cells and plays a crucial role in the maintenance of the self-renewal capacity of these cells. ▪ Is involved in the early neural differentiation and maturation process. ▪ Regulates cell differentiation and proliferation. ▪ Induces early neuronal differentiation.	[29,55,58,59]
**GD2**	▪ Is a marker for neuronal differentiation. ▪ Is a promoter for the differentiation of neuronal cells	[26,55,60]
**GD1a**	▪ Is prominently expressed in neurons and induces early neural differentiation. ▪ It enhances EGF-induced EGFR phosphorylation, which promotes osteoblast differentiation. ▪ Regulates the initial steps of osteoblast differentiation. ▪ Is important for beta-glycophosphate, ascorbic acid, and dexamethasone induced osteoblastogenesis.	[28,29,47,61,62]
**GD1b**	▪ It is expressed after neuronal differentiation with consequent loss of the “stemness” of NSCs.	[63]
**GT3, GT1c, GQ1c**	▪ Are used for isolating embryonic neural stem cells from rat telencephalon and to label neuroglial progenitor cells and O-2A progenitor cells.	[64]
**GT1b**	▪ Induces differentiation of mESCs and MSCs into neuronal cells. ▪ Is a biomarker to neuronal differentiation.	[55,65]
**GQ1b**	▪ Promotes neurite outgrowth during early neural differentiation via the ERK1/2 MAPK pathway.	[65,66]

**Table 2 biomedicines-10-03112-t002:** Change in gangliosides pattern during multilineage differentiation of MSCs.

Gangliosides Expression						
	Undifferentiated	Osteogenic Differentiation	Chondrogenic Differentiation	Neurogenic Differentiation	Adipogenic Differentiation	Ref
**UC-MSCs**	GM3 GM1 GD2	n.i.	n.i.	n.i. n.i. GD2	n.i.	[80,84,85]
**BM-MSCs**	GM3 GM2 GM1 GD3 GD2 GD1a	GM3 ↓ GM2 ↑ GM1 ↑ GD3 ↓ n.e. GD1a ↑↑	GM3 ↑ ↓ n.e. GM1 ↑ GD3 ↑ n.i. GD1a ↑	GM3 n.e. GM1 n.i. n.e. n.i.	n.i.	[26,28,29,35,37,47,52,80,84,85,87,88,89]
**DPSCs**	GM3 GM2 n.e. GD1a	GM3 ↑ GM2 ↑ n.e. GD1a ↑↑	n.i.	GM3 GM2 ↓ GD3 ↑ GD1a ↑	n.i.	[4,5,28,29,35,36,45,47]
**ADSCs**	GM3 GM2 GM1 GD3 GD2 GD1a	GM3 ↑ GM2 ↑ GM1 ↑ GD3 ↑↑ n.e. GD1a ↑	GM3 ↑ GM2 ↑ GM1 ↓ GD3 ↓ n.e. n.e.	n.i. n.i. n.i. GD3 n.i. n.i.	GM3 ↑↑ GM2 ↑↑ GM1 ↑ GD3 ↑ GD2 ↓ GD1a ↑	[26,36,45,47,52,54,90,91,92]

n.i. = Not investigated, n.e. = Not expressed, ↑ Increase expression, ↓ Decrease expression.

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
