# Peer review of "Gangliosides and Their Role in Multilineage Differentiation of Mesenchymal Stem Cells"

_biomedicines, 2022, doi:10.3390/biomedicines10123112_

Round 1

Reviewer 1 Report

The review "Gangliosides and their role in multilineage differentiation of mesenchymal stem cells" by Francesca Santilli , Jessica Fabrizi , Fanny Pulcini , Maurizio Sorice , Costantino Santacroce , Simona Delle Monache , Vincenzo Mattei is about the role of gangliosides as specific markers of the cellular state of undifferentiated and differentiated MSCs, towards the osteogenic, chondrogenic, neurogenic and adipogenic lineage. it is a vary interesting and novel topic and the review is well written. However I have some questions and requests for the authors:

The authors could write a short paragraph to the classification and biochemical description of gangliodids.

Do MSC gangliosides play a role in the microenvironment? Are they important to support hematopoiesis? Are described MSC / CD34 co-culture assays in which the phenotype of MSC gangliosides changed ? Please clarify this aspect.

Can the authors schematize the variation of the ganglioside pattern during adipogenic, osteoblastic and neural differentiation?

Are gangliosides expressed on MSCs isolated from the nose or nasal polyp lesions?

The authors could write a short paragraph about the classification and biochemical description of gangliosides.

Gangliosides are differentially expressed in normal and malignant MSCs? 

Author Response

- The authors could write a short paragraph to the classification and biochemical description of gangliosides.
Answer: A brief description of gangliosides has been added in introduction. Moreover, a schematic representation of gangliosides and metabolic pathways of ganglio-series glycosphingolipids has been added in Figure 1.

See Page 2 from line 44 to line 52 and page 45 from line 1 to line 9

See Figure 1 page 3 

- Do MSC gangliosides play a role in the microenvironment? Are they important to support hematopoiesis? Are described MSC / CD34 co-culture assays in which the phenotype of MSC gangliosides changed? Please clarify this aspect.

Answer: As suggested by reviewer several authors demonstrated a role of MSCs gangliosides in tumor microenvironment. Moreover, gangliosides shed by tumor cells modulate hematopoiesis and contribute to human tumor-associated suppression of hematopoiesis. In addition, since it has been reported that hematopoietic cells do not express the ganglioside it is possible using them as markers of MSCs.

                We added a sentence in the Introduction page 3 from line 14 to line 24

                Page 2 from line 41 to line 43

- Can the authors schematize the variation of the ganglioside pattern during adipogenic, osteoblastic and neural differentiation?
Answer: Indeed, we summarized the variation of the ganglioside pattern during adipogenic, osteoblastic and neural differentiation by means of the Table 2 because we thought that it is the most efficient way to describe the process. In fact, the description of the expression pattern of 4 types of mesenchymal stem cells in multilineage differentiation process by wording, even if schematic or brief could not be meaningful nor easy to understand.

See Page 6

- Are gangliosides expressed on MSCs isolated from the nose or nasal polyp lesions?
Answer: We did a revision of scientific literature on PubMed but unfortunately, we have not found information about the expression of gangliosides on MSC isolated from the nose or nasal polyp lesions.

Reviewer 2 Report

The topic seems interesting, however, the lack of literature on it.

The authors are requested to address the following points:

1-The authors are suggested to briefly describe the expression and the function of gangliosides in the pluripotent stem cells and based on that, the authors need to give an overview of applications of that knowledge in multipotent MSCs in the future studies.

2-Is there any correlation between gangliosides expression and tumorigenicity of MSCs?

3-The authors are need o discuss how to apply these markers in evaluation of stem cells quality.

4-I there any relationships between the expressions of gangliosides and stem cells migration, adhesion, and proliferation?

5- The prepared tables and figure are simple and need further elaboration

Author Response

The topic seems interesting, however, the lack of literature on it.

The authors are requested to address the following points:

1-The authors are suggested to briefly describe the expression and the function of gangliosides in the pluripotent stem cells and based on that, the authors need to give an overview of applications of that knowledge in multipotent MSCs in the future studies.

Answer: We added a sentence in the paragraph of Introduction remarking the role of ganglioside in the pluripotent stem cells.

See Page 2 from line 33 to line 36 

See Page 12 from line 18 to line 20

Regarding future application of gangliosides in multipotent MSCs we added a short paragraph in conclusion and future perspectives.

See Page 10 from line 25 to line 28 and page 11 from line 1 to line 5

2-Is there any correlation between gangliosides expression and tumorigenicity of MSCs?

Answer: Yes, a strong correlation between gangliosides expression and tumorigenicity of MSCs in several kind of tumor has been demonstrated. We added some literature studies about this correlation:

We added a sentence in the Introduction page 3 from line 14 to line 24

3-The authors are need to discuss how to apply these markers in evaluation of stem cells quality.

Answer: Our article focused on the possibility to use gangliosides as markers of mesenchymal stem cell differentiation. To this end GGs could effectively be used to do a selection of phenotypically and functionally homogenous cells. These cells should be more capable of expansion and self-renewal. Therefore, selecting this subset of cells is expected to obtain a high-quality stem cell population.

We reported it on page 2 from line 33 to line 36 and page 12 from line 18 to line 20

4-I there any relationships between the expressions of gangliosides and stem cells migration, adhesion, and proliferation?

Answer: Yes, this relation has been reported in our review in all paragraphs regarding UC-MSCs, BM-MSCs, DP-MSCs and AD-MSCs GGs expression.

5- The prepared tables and figure are simple and need further elaboration

Answer: We improved table 1 with missing data and standardized table 2. We have also slightly modified Figure 2. We also added the figure 1 where we show a schematic representation of gangliosides metabolism of ganglio-series glycosphingolipids.

See Table 1 page 4-5, Table 2 page 6 and Figure 1 page 3

Reviewer 3 Report

This is quite an interesting review article which is short and crisp. I enjoyed reading it. However, I believe that the revisions as per the below suggestions could make it in a better format.

1.     Manuscript needs proper english editing to remove grammatical errors. For instance, in Abstract, it should be “neurogenic, and adipogenic lineage”.

2.     It is not appropriate to use a keyword like “gangliosides as markers of stem cells”.

3.     Table 1 is not exhaustive. Information about some gangliosides are missing. For instance, GD1b.

4.     Formatting in Table 2 should be uniform. Somewhere, it is centralized, somewhere it is left aligned.

5.     Limitation of the present review article should also be highlighted by the authors.

6.     Further, there should be a new section, as “Future perspectives”.

7.     Role of different MSCs as a regenerative medicine could be better described by a suitable figure. 

Author Response

This is quite an interesting review article which is short and crisp. I enjoyed reading it. However, I believe that the revisions as per the below suggestions could make it in a better format. 

  1. Manuscript needs proper english editing to remove grammatical errors. For instance, in Abstract, it should be “neurogenic, and adipogenic lineage”.

Answer: Thanks for the advice. A correction was made by a native speaker.

  1. It is not appropriate to use a keyword like “gangliosides as markers of stem cells”.

Answer: According to suggestion referee keywords have been modified.

  1. Table 1 Is not exhaustive. Information about some gangliosides are missing. For instance, GD1b.

Answer: Table 1 has been improved.

See Table 1 page 4-5

  1. Formatting in Table 2 should be uniform. Somewhere, it is centralized, somewhere it is left aligned.

Answer: Table 2 has been uniformed.

See Table 2 page 6

  1. Limitation of the present review article should also be highlighted by the authors.

Answer: The main limitation is the lack of scientific literature on the subject. We have inserted a sentence in the “Conclusions and Future perspectives”.

See page 11 from line 7 to line 9

  1. Further, there should be a new section, as “Future perspectives”.

Answer: A new section has been added

See page 10-11

  1. Role of different MSCs as a regenerative medicine could be better described by a suitable figure. 

Answer: Scientific literature nowadays offers a multitude of figures describing the role of MSC, therefore we judge redundant and even off topic to add such figure in this review, which is pointing more in depth on the use of gangliosides as stem cell markers.

Round 2

Reviewer 1 Report

The authors answered my questions and significantly improved the manuscript. The review can be considered publishable on Biomedicines.

Reviewer 2 Report

I recommned the acceptance.